# Cognitive Personalized Search Integrating Large Language Models with an Efficient Memory Mechanism

## ABSTRACT

Traditional search engines usually provide identical search results for all users, overlooking individual preferences. To counter this limitation, personalized search has been developed to re-rank results based on user preferences derived from query logs. Deep learning-based personalized search methods have shown promise, but they rely heavily on abundant training data, making them susceptible to data sparsity challenges. This paper proposes a Cognitive Personalized Search (CoPS) model, which integrates Large Language Models (LLMs) with a cognitive memory mechanism inspired by human cognition. CoPS employs LLMs to enhance user modeling and user search experience. The cognitive memory mechanism comprises sensory memory for quick sensory responses, working memory for sophisticated cognitive responses, and long-term memory for storing vast historical interactions. CoPS effectively handles new queries using a three-step approach: identifying re-finding behaviors, constructing a user profile with relevant historical information, and ranking documents based on personalized query intent. Experimental results demonstrate the superiority of CoPS over baseline models in zero-shot scenarios.

## KEYWORDS

Personalized Search, Large Language Models, Memory Mechanism, Cognitive Psychology

## 1 INTRODUCTION

Search engines have become indispensable tools for information retrieval and are ubiquitously used across the globe. However, traditional search engines often employ a one-size-fits-all approach, delivering identical search results for a given query regardless of the diverse need of individual users. Such an approach overlooks the varied interests and preferences of individual users. To address this gap, personalized search has been developed as a strategy to re-rank search results, catering to each user's distinct preferences [27].

Personalized search primarily involves modeling user preferences by analyzing their query logs and past interactions. Initial efforts in this domain focused on extracting features from user click-through data to predict interests [2]. The landscape shifted with the advent of deep learning-based methods [11], which construct user profiles in a semantic space, substantially improving search

*Conference'17, July 2017, Washington, DC, USA*
© 2023 Association for Computing Machinery.
ACM ISBN 978-x-xxxx-xxxx-x/YY/MM...$15.00
https://doi.org/10.1145/nnnnnnn.nnnnnnn

performance. However, a critical limitation of these deep learning approaches is their dependence on vast amounts of training data [37], which are comprised of data from many users and may bring challenges to user privacy protection. This reliance creates a significant bottleneck, particularly in situations where only limited high-quality individual data is available for model training. Large Language Models (LLMs), however, offer a promising solution to this problem. Known for their exceptional performance in zero-shot contexts, LLMs can perform complex tasks without the need for task-specific fine-tuning or vast amounts of training data. In light of this, we propose to refine user modeling by integrating LLMs' ability to work effectively in zero-shot scenarios.

However, directly applying LLMs to personalized search introduces its own set of challenges. Specifically, when deploying LLMs for personalized search tasks, the long-range nature of user histories can become an obstacle. These histories, containing user interactions, are critical to creating tailored search results. However, as these histories grow in length and complexity, processing such extensive data can be computationally intensive and even exceeds the length limit of LLMs. To tackle this issue, we propose the construction of an LLM-based personalized model with external memory units, ensuring that the LLM can rapidly access the most relevant segments of the history without processing the entire user history. Moreover, as the user history grows, the external memory can be dynamically expanded by encoding user interactions at the end of each session. This ensures that the system remains scalability and can handle increasing amounts of data without significant drops in efficiency.

To further boost the performance of the LLM-based personalized search model, we have taken inspiration from one of the most sophisticated processing units known to mankind - the human brain. Intriguingly, despite the vast and often noisy information the brain holds, humans are able to respond swiftly and accurately to external stimuli [16]. This cognitive proficiency mirrors the challenge posed by user histories in personalized search – extensive, multifaceted, and noisy. Therefore, it stands to reason that imitating the cognitive memory mechanism of the human brain could offer benefits when constructing the external memory units for our model.

According to recent findings of cognitive psychology [5, 23], the memory mechanism of the human brain is segmented into different components - sensory memory, working memory, and long-term memory, as depicted in Figure 1. Sensory memory is the earliest stage of memory, holding sensory information and facilitating rapid response to stimuli instantaneously. Once the information has passed through the sensory memory, it progresses to the working memory, which integrates new information with existing knowledge retrieved from long-term memory. At last, information from working memory is encoded to long-term memory, from where it can be retrieved when needed. Long-term memory stores enduring information encompassing knowledge and experiences. Together,

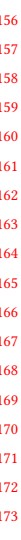

**Figure 1: The memory mechanism of the human brain.**

these modules form a system that efficiently processes, retains, and retrieves information.

Drawing parallels with the cognitive memory mechanism, we construct our external memory units with a similar structure. Designed for swift query processing, the sensory memory unit identifies if a query relates to a re-finding behavior—essentially revisiting previously accessed content. Recognized re-finding queries are instantly ranked while others are sent to the working memory for deeper analysis. The working memory unit assesses the query against the user's recent history and collaborates with the long-term memory to integrate past user interests. These data form a user profile which the LLM uses to model user intent. Serving as a vast store of user preferences, the long-term memory aids the working memory by providing deep insights into the user's long-term interests and habits.

In this paper, we propose a Cognitive Personalized Search model (CoPS), which leverages the strengths of LLMs and integrates cognitive memory mechanisms to optimize user modeling during the search process. CoPS incorporates three essential components to facilitate personalized search results: (i) a cognitive memory mechanism as the central storage unit, (ii) an LLM as the central cognitive unit, and (iii) a ranker as the central scoring unit. Specifically, the CoPS employs a three-step approach to effectively handle new queries. Firstly, CoPS evaluates whether a new query corresponds to a re-finding behavior using its sensory memory. If identified as re-finding action, CoPS instantly utilizes the sensory response to rank documents. Otherwise, the query is forwarded to the working memory for further analysis. In the second step, the working memory integrates relevant historical information, encompassing the user's short-term history and query-aware user interests retrieved from long-term memory, to construct a foundational user profile for user modeling by the LLM. Lastly, CoPS employs a ranking component to prioritize the candidate documents based on the user's personalized query intent. Experimental results on two datasets demonstrate that our proposed model outperforms baseline models in zero-settings.

Our contributions are summarized as: (1) We propose an LLM-empowered cognitive personalized search model that incorporates LLMs to improve user modeling. (2) We integrate the external memory units with LLM to provide efficient access to extensive user histories. (3) We organize the memory units with an architecture that imitates the memory mechanism of the human brain, ensuring the model scalability and performance with large amounts of data.

## 2 RELATED WORK

### 2.1 Personalized Search Models

Personalized search has gained considerable attention due to its effectiveness in providing satisfactory results tailored to individual users [3]. Traditional approaches often rely on heuristic rules or manually extracted features to implement personalized search [1, 4, 8, 12, 30]. A recent shift has seen the rise of deep learning-based techniques in personalized search, demonstrating superior capabilities in learning implicit user interests [11, 29, 32, 34–36]. Ge et al. [11] have employed RNN structures to leverage sequential data to craft detailed user profiles. To improve the representation of user profiles and mine high-quality negative samples, GAN-based models have been proposed [15]. Memory networks have been employed to capture multi-level re-finding behavior in personalization [36]. Contextual information has been incorporated to learn clear query representations [32, 35]. More recently, a multi-task contrastive learning model has been developed, achieving better performance in personalized search [37]. However, these neural personalized search approaches rely heavily on abundant training data. In response to this challenge, our paper presents an LLM-empowered personalized search framework that enables user modeling in zero-shot scenarios.

### 2.2 LLMs in Information Retrieval

The emergence of large language models has significantly advanced information retrieval, particularly in the domains of document ranking and personalized recommendation tasks. In the context of document ranking, several studies [19, 21, 25, 26] have explored how to leverage large language models to match queries with documents, employing different approaches such as pairwise and listwise methods. These approaches aim to optimize the ranking of documents based on their relevance to a given query. In the realm of personalized recommendation, researchers [6, 9, 10, 13, 14, 28] have investigated the potential of large language models in extracting user interests through techniques like prompt designing and in-context learning. These efforts have focused on harnessing the capabilities of large models to enhance the accuracy and effectiveness of personalized recommendation systems. Different from previous tasks, this paper conducts a comprehensive investigation into the integration of LLMs into personalized search systems, aiming to advance the utilization of LLMs for handling personal data.

## 3 COGNITIVE PERSONALIZED SEARCH

Personalized search has emerged as an effective approach to enhance the user search experience through the modeling of user interests. However, as mentioned previously, existing personalization models encounter significant challenges stemming from data sparsity and the complexities associated with user history. To address these limitations, this paper introduces a framework empowered by large language models with an efficient memory mechanism to enhance user modeling in personalized search.

To begin with, the task of personalized search can be defined as follows. The user's historical data, represented as $H$, is comprised of both short-term history, denoted as $H^s = \{I_1^s, \cdots, I_{t-1}^s\}$, capturing the current session's sequence of user interactions, and long-term history, denoted as $H^l = \{I_1^l, I_2^l, \cdots\}$, encompassing past interactions from previous sessions. Each interaction $I_i$ is recorded in the search log and comprises a user-issued query, skipped documents, and clicked documents, denoted as $\{q_i, D_i^-, D_i^+\}$. Given a new query $q$ and a set of candidate documents $D = \{d_1, d_2, ...\}$ retrieved by the

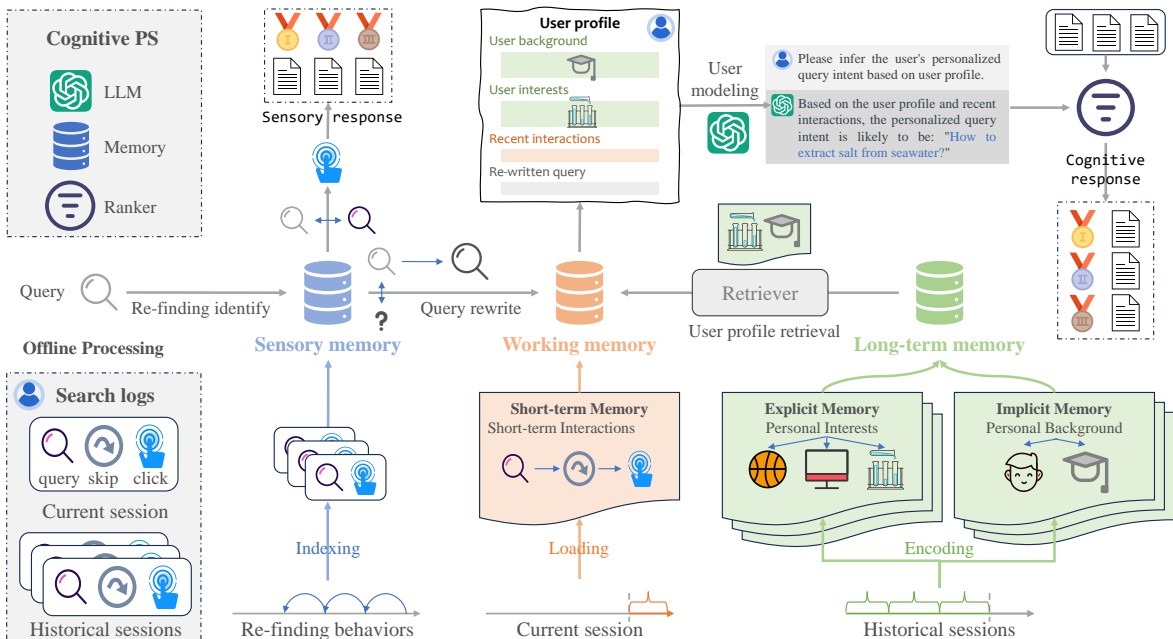

**Figure 2: The overview of CoPS. The system initially engages the sensory memory to identify re-finding behaviors, thus generating a sensory response if identified. Otherwise, the working memory collaborates with an LLM to accumulate personalized cues related to the query. After LLM-empowered user modeling, a ranker is employed to re-rank the results based on user interests. Note that CoPS does not require any training data and personalization can be achieved with the user's own data.**

search engine, the primary objective of personalized search is to determine a score, $p(d|q, H)$, for each document in $D$, taking into account both the current query $q$ and the historical data $H$.

As shown in Figure 2, we develop a cognitive personalized search model empowered by LLMs to calculate the above personalized scores. We elaborate our model based on three main components as follows: (1) Memory mechanism as a central storage unit, (2) LLM as a central cognitive unit, and (3) Ranker as a central scoring unit.

## 3.1 Memory Mechanism: Central Storage Unit

Due to the extensive nature of user histories, we propose to leverage an external cognitive memory as the central storage unit. As outlined previously, the human brain exhibits a sophisticated cognitive memory structure, comprising a sensory memory, a working memory, and a long-term memory. This structure ensures swift and effective reactions to external stimuli. In mirroring these capabilities of the human brain, we endeavor to integrate these memory units into LLMs, enabling personalized storage of user interactions and efficient feedback mechanisms.

*3.1.1 Sensory Memory.* The primary function of sensory memory lies in its ability to provide immediate feedback for external stimuli. In the context of personalized search, a user behavior pattern known as "re-finding" has been observed in [8]. This pattern emerges when users search for information that they have previously encountered, making it a simple but effective way to predict a user's next clicks in personalized search. Inspired by this observation, we propose the idea of archiving user re-finding behaviors within the sensory

memory. This approach allows for the swift identification of re-finding instances and promotes the generation of an immediate sensory reaction.

Specifically, CoPS extracts all pairs of query and clicked document, denoted as $(q, d^+)$, from the user's historical data $H$, whereby the frequency of each document being clicked is recorded. Upon receiving a new query, it first prompts the sensory memory for feedback. If a match is found within the sensory memory, the query is classified as a re-finding behavior. CoPS then produces a sensory response by ranking the candidate documents based on their click frequency. If no match is found, the query is forwarded to the working memory for further processing and analysis.

*3.1.2 Working Memory.* Working memory serves as a crucial cognitive system, responsible for the temporary storage and integration of information related to ongoing tasks. In the context of personalized search, the user's personalized query intent can be effectively captured by considering three crucial dimensions of information:

- *Relevant interactions.* The user's historical search behavior often contains noisy and irrelevent data for personalization. Therefore, the information directly related to the current query within the user's search history takes on greater significance in tailoring search results.
- *Contextual interactions.* Users often enter a series of queries within a single session to satisfy a particular information need. These queries, coupled with the corresponding skip and click behaviors in the current session $H^s$, provide rich contextual clues.

Utilizing this information can significantly enhance the model's ability to infer the user's present intent.

- *Re-written Query.* Queries formulated by users are typically brief and may contain typos or other inconsistencies, potentially hindering the accurate interpretation of the underlying intent. The refinement or re-writing of these queries is thus a key step in better understanding and responding to the user's specific information needs.

*3.1.3 Long-term Memory.* Long-term Memory plays a pivotal role in memory systems, enabling the retention of user-specific signals over extended temporal spans. In personalized search, long-term memory is primarily devised for the preservation of the user's long-term interactions. However, due to the vastness of user history and the presence of a substantial amount of noisy data, it becomes necessary to segment and encode the long-term history $H^l$, retaining the most salient personalized signals. Concretely, we partition the user history into fixed-length temporal windows. Interactions within these intervals are allocated to specific memory slots. To holistically capture users' personalized signals, we encode user interactions from two dimensions: explicit and implicit memory.

- *Explicit Memory.* The explicit memory primarily captures the user's specific interests and preferences within each topic, serving as a valuable resource for tailoring candidate documents to individual needs. For each interaction window, data is encoded into a key-value storage format: the key represents the topic, while the value details the items the user showed interest in, within that topic. To implement this, we employ the LLM with several demonstrations and the instruction "*[Demonstrations] [User interactions]. Please summarize the user interests into multiple topics based on the user's historical query log*".
- *Implicit Memory.* Implicit memory focuses on storing personal contextual details such as occupation, gender, and other underlying latent factors. Together, these elements contribute to discerning a user's personalized query intentions. For instance, the appearance of words like "python" in the query might suggest the user's occupation as a programmer. Similar to the explicit memory, we employ the LLM for the encoding process. It's worth noting that the encoding method here can be replaced by alternative document summarization models or vector encoding models.

With our system now possessing memory capabilities, there remains an essential need for a cognitive module to handle information extraction and analysis. The LLM shown below is designed for this role.

## 3.2 LLM: Central Cognitive Unit

Large language models have emerged as revolutionary breakthroughs in the realm of natural language processing. Their exceptional skill in comprehending and generating human language, especially in areas like common-sense reasoning and knowledge extraction in zero-shot contexts, underscores their significance. These abilities qualify the LLM as a key cognitive component in personalized search, where it can significantly improve user modeling and interpret personalized query intentions. Specifically, the LLM in CoPS determines the information to be loaded into working memory and executes cognitive reasoning to analyze the contents therein, encompassing tasks such as query re-writing, user profile retrieval, and user interest modeling.

*3.2.1 Query Re-writing.* Query re-writing involves refining and transforming user queries to improve their clarity. Given this, LLMs can decipher the initial user query and its deeper meaning, suggesting alternative phrasings that may fit the context better. By utilizing query rewriting, CoPS stands a better chance of identifying relevant historical interactions and accurately modeling a user's personalized query intent. Specifically, when given a query, the LLM is leveraged to perform query re-writing with a designated prompt, such as:

- *[Query]. Please act as a query re-writer to enrich the query and make the query intent clearer.*

*3.2.2 User Profile Retrieval.* User profile retrieval enables the retrieval and utilization of information from the long-term memory. The LLM acts as a pivotal tool in accessing relevant user profiles from the long-term memory based on the current query intent. To access the explicit and implicit memories, the LLM is prompted in a manner specifically tailored to extract relevant information from each memory slot in relation to the current query, such as:

- *[An Explicit/Implicit memory slot], [Re-written query]. Please act as a retriever to extract personal interests/backgrounds related to the query from the memory.*

The results of the rewriting and retrieval processes, including the refined query and the retrieved user profiles, are stored in the working memory. This storage also considers the user's recent interactions, which aids in future user modeling. It's worth noting that the choice of retriever here can be swapped with traditional sparse or dense retrievers. This allows for the decoupling of the LLM from external memory.

*3.2.3 User Modeling.* User modeling aims to understand and predict user behavior and preferences. In our cognitive personalized search model, user modeling primarily involves processing the information stored in working memory, including rewritten queries, short-term user interactions, and retrieved user profiles to infer the user's personalized query intent. To do this, the LLM is given prompts such as:

- *[User background], [User interests], [Recent Interactions], [Re-written Query]. Please infer the user's personalized query intent based on the user profile.*

The output of the LLM is then considered as the query-aware user preferences, denoted by $U_{q,H}$. In contrast to previous user modeling methods reliant on deep learning, a notable advantage of LLM-empowered user modeling lies in its representation of user interests through natural language rather than highly abstract vectors. This approach grants the model a higher level of interpretability.

## 3.3 Ranker: Central Scoring Unit

The ranker serves as the central scoring unit, determining the relevance and order of search results. In CoPS, the ranker is specifically designed to weigh the correlation between user preferences and document content, thus prioritizing documents that resonate with user inclinations. Formally, the personalized ranking score for a

document $d$ is represented as: $p(d|q, H) = \mathcal{R}(U_{q,H}, d)$, where $\mathcal{R}$ denotes the ranking function. We explore three distinct types of rankers within zero-shot settings:

*3.3.1 Term-based Ranker.* The term-based ranker operates by analyzing the frequency and distribution of query terms within candidate documents. It relies on established term-matching techniques to evaluate the relevance of documents to the user's query. Due to its computational efficiency and straightforward deployment, this method has become a commonly employed approach in information retrieval systems. In CoPS, since user preferences are expressed in natural language, the BM25 model [22] is adopted to compute a personalized score for each document.

*3.3.2 Vector-based Ranker.* The vector-based ranker comprehends contextual information and interrelationships among words in a sentence. This allows it to grasp the contextual relevance of candidate documents in relation to the query. In CoPS, we adopt an interaction-based BERT ranker, which concatenates the user preferences $U_{q,H}$ and the document $d$ with the '[SEP]' token as the separator. Subsequently, this combined representation is fed into a pre-trained BERT model, and the matching score is computed using a linear layer. Specifically, we choose DistilBERT [24], which is trained on the MS MARCO dataset [17] for the ranking task, to compute the matching score.

*3.3.3 LLM-based Ranker.* The LLM-based ranker presents a new ranking paradigm that directly generates the ranking list for a given query and candidate documents [26]. In CoPS, we input the user preferences and candidate documents into the LLM, then request a personalized ranking list using the following prompt:

- *[Query], [User preferences], [Candidate documents]. Please rank these documents by measuring the relevance based on the query and user preferences.*

In summary, we introduce CoPS, a fusion of LLMs and a cognitive memory mechanism aimed at elevating user modeling to enhance personalized search results. Unlike prior deep learning approaches, our method operates in an unsupervised manner throughout the entire process of search result personalization. This design aims to amplify personalization even in scenarios where no training data is available, and is conducive to privacy protection.

## 3.4 Discussions

As the fusion of LLMs with personalization progresses, we delve into the primary challenges concerning LLM's role in facilitating personalized search:

- **How can user log data inputted through an online interface into an LLM be safeguarded to protect user privacy?**

Uploading individual search data to a closed-source LLM through an online interface may compromise user privacy. A viable solution is to deploy open-source models (like the vicuna model used in section 5.3) on local devices. Additionally, to avoid utilizing private user data for model training, we propose a framework called CoPS. This framework leverages external memory mechanisms and retrieval techniques, feeding only a minimal amount of retrieved user-relevant data into the LLM for inference, thus achieving accurate user modeling without model training.

- **How to effectively achieve low-latency search result delivery in such a LLM-based system?**

Over time, the accumulation of user query logs will grow, significantly slowing down the LLM's response speed, which in turn affects the user experience. To mitigate this issue, our CoPS framework utilizes a cognitive memory mechanism to accelerate the process from two fronts: Firstly, handling simple repetitive queries using sensory memory will expedite response times. Secondly, employing retrieval techniques to feed only the current query-relevant user interests into the LLM for user modeling could further enhance the speed and efficiency of personalized search responses.

By addressing the privacy concerns and ensuring swift response times, it enhances the feasibility and user satisfaction in deploying LLMs for personalized searches. Through local model deployment and efficient memory and retrieval mechanisms, we can stride towards a more user-centric and privacy-compliant personalized search experience.

## 4 EXPERIMENTAL SETTINGS

### 4.1 Datasets and Evaluation Metrics

Due to the scarcity of datasets suitable for personalized search, we carefully selected the following two datasets for experimentation: the AOL search logs dataset [18] and a commercial dataset obtained from a large-scale search engine. Each piece of data includes an anonymous user ID, a session ID, a query, the timestamp, a document, and a binary click tag. For our experimentation, we used the query logs from the first 85% interactions to represent the user history, while the queries issued in the subsequent 15% interactions for model testing. To manage the expenses of invoking the LLM API, we strategically sampled 200 users with massive interactions from each dataset, including 16,626 and 10,294 queries respectively.

In evaluating the performance of our model, we utilize several commonly used metrics, including MAP, MRR, and precision (P@1), to assess the quality of the ranking. In addition, following [11], we adopt an additional metric, P-improve (P-imp), to measure the reliable improvements on inverse document pairs.

### 4.2 Baselines

For comparison, our study incorporates a diverse set of baselines representing both ad-hoc and personalized search models. In each category, we carefully selected both fine-tuned models and zero-shot models to test the performance of the model with or without sufficient training data.

**KNRM** [31]. Tailored for ad-hoc search applications, KNRM leverages kernel-pooling to generate multi-level soft matching features from a word similarity matrix, establishing a nuanced ranking framework.

**Conv-KNRM** [7]. Building upon KNRM, Conv-KNRM incorporates a convolutional layer to model n-gram soft matches, harnessing the contextual essence of surrounding words to enhance the precision of matching.

**BERT** [20]. Engaging the pre-trained BERT model, this approach tackles the query-document matching challenge. By concatenating query-document sequences and channeling them through the BERT model, the representation of the [CLS] token in the final layer is adopted as the matching feature.

**Table 1: Overall performance of all models on two datasets. Zero-shot represents whether the model can be applied to zero-shot scenarios. "†" indicates the model outperforms zero-shot baselines significantly with paired t-test at p < 0.05 level.**

| Task | Model | Zero-shot | AOL dataset | | | | Commercial dataset | | | |
|------|-------|-----------|------|------|------|-------|------|------|------|-------|
| | | | MAP | MRR | P@1 | P-imp | MAP | MRR | P@1 | P-imp |
| Adhoc Search | KNRM | - | .4291 | .4391 | .2704 | .3634 | .4916 | .5001 | .2849 | .0655 |
| | Conv-KNRM | - | .4738 | .4849 | .3266 | .4293 | .5872 | .5977 | .4188 | .1422 |
| | BERT | - | .5033 | .5135 | .3552 | .6082 | .6232 | .6326 | .4475 | .1778 |
| | BM25 | ✓ | .3617 | .3717 | .2549 | .2710 | .4702 | .4808 | .2682 | .1484 |
| | DistilBERT | ✓ | .3762 | .3811 | .2383 | .5148 | .4154 | .4160 | .1972 | .2301 |
| | ChatGPT | ✓ | .5082 | .5122 | .3731 | .5142 | .6023 | .6379 | .4413 | .1886 |
| Personalized Search | SLTB | - | .5113 | .5237 | .4693 | .3374 | .7023 | .7104 | .6105 | .1398 |
| | HRNN | - | .5324 | .5545 | .4854 | .5927 | .8065 | .8191 | .7127 | .2404 |
| | RPMN | - | .5926 | .6049 | .5322 | .6586 | .8238 | .8342 | .7305 | .2652 |
| | HTPS | - | .7091 | .7251 | .6268 | .7730 | .8222 | .8324 | .7291 | .2554 |
| | P-Click | ✓ | .4221 | .4305 | .3780 | .1657 | .6802 | .6935 | .5668 | .0625 |
| | CoPS (Ours) | ✓ | .7043† | .7081† | .5906† | .7229† | .8018† | .8153† | .7353† | .3008† |

**Table 2: The results with different memory units. ● and ○ indicates the model with and without the memory.**

| Model | Memory Units | | | | AOL dataset | | | Commercial dataset | | |
|-------|--------------|---------|--------|--------|------|---------|-----------|------|---------|-----------|
| | Sensory | Working | Long-E | Long-I | MAP | | Latency (s) | MAP | | Latency (s) |
| CoPS | ○ | ● | ● | ● | .6863 | ↓ 2.55% | 1.13 ×2.09 | .7641 | ↓ 2.26% | 1.35 ×2.17 |
| CoPS | ● | ○ | ● | ● | .6582 | ↓ 6.54% | 0.43 ×0.79 | .7413 | ↓ 5.18% | 0.50 ×0.81 |
| CoPS | ● | ● | ○ | ● | .6281 | ↓ 10.8% | 0.27 ×0.50 | .7326 | ↓ 6.29% | 0.33 ×0.53 |
| CoPS | ● | ● | ● | ○ | .6807 | ↓ 8.05% | 0.28 ×0.52 | .7700 | ↓ 1.51% | 0.35 ×0.56 |
| CoPS | ● | ● | ● | ● | .7043 | - | 0.54 - | .7818 | - | 0.62 - |

**Table 3: The role of LLM in query re-writing (QR), user profile retrieval (UPR) and user modeling (UM).**

| Model | AOL dataset | | Commercial dataset | |
|-------|------|------|------|------|
| | MAP | MRR | MAP | MRR |
| CoPS-ChatGPT | .7043 | .7081 | .7818 | .8153 |
| QR-Remove | .6872 | .6898 | .7702 | .7923 |
| QR-Vicuna | .6903 | .6963 | .7731 | .7971 |
| UPR-Random | .6103 | .6222 | .7189 | .7408 |
| UPR-Vicuna | .6382 | .6483 | .7217 | .7492 |
| UM-Remove | .6217 | .6290 | .7266 | .7554 |
| UM-Vicuna | .6423 | .6507 | .7362 | .7601 |

**BM25** [20]. This method computes the lexical-level relevance between queries and documents based on IF-IDF weighting.

**DistilBERT** [24]. Utilizing a large-scale query-document relevance dataset, MS MARCO, this model is trained specifically for ranking tasks, demonstrating strong generalization capabilities.

**ChatGPT** [26]. By designing prompts, this model manages to directly leverage LLM to output ranking results.

**SLTB** [2]. It employs a comprehensive feature aggregation strategy by combining click features, topical features, time-related features, and positional features to enhance result personalization through a learning-to-rank methodology.

**HRNN** [11]. Focused on personalized search, HRNN employs sequential analysis of query logs to construct dynamic user profiles based on the current query. By integrating hierarchical recurrent neural networks with query-aware attention mechanisms, it effectively embodies this concept.

**RPMN** [36]. This memory network-centric personalized search model seeks to unearth potential re-finding behaviors. By crafting three external memories, it adeptly navigates through different types of re-finding behavior.

**HTPS** [35]. Structured as a personalized search framework, HTPS employs a hierarchical transformer to initially encode history, facilitating query disambiguation through contextual information.

**P-Click** [8]. This model leverages the frequency of clicks on a particular document and its original position to effectively re-rank search results using the Borda count method. P-Click is designed to prioritize the user's behavior when revisiting or re-finding information, making it a robust approach for enhancing search results.

## 4.3 Implementation Details

For all fine-tuned baseline models, the training process is conducted using the complete set of interactions contained within the training dataset, while our model requires no training data. The LLM is accessed through OpenAI's API, specifically the `gpt-3.5-turbo` variant. The temperature parameter for calling LLMs is set to 0.2, a value that balances trade-offs between model uncertainty and

response variability. For ranker selection, the ChatGPT [26] ranker is employed for both datasets.

## 5 EXPERIMENTAL RESULTS

This section presents the experimental results of our proposed approach and conducts an empirical analysis to offer a comprehensive understanding of the results.

### 5.1 Overall Performance

The overall results of models are displayed in Table 1. We can observe that:

(1) CoPS outperforms existing zero-shot personalized search methods on both datasets, demonstrating consistent and robust capabilities in effectively modeling users without dependency on prior specific training. In comparison with the zero-shot baseline P-Click, which also leverages user re-finding behaviors for search result personalization, CoPS achieves a significant edge with improvements of 66.8% and 14.9% in MAP metrics for the two datasets respectively. These results underscore CoPS's proficiency in integrating information from both working memory and long-term memory through the LLM. As a result, CoPS is able to identify and mine user interests, even when queries do not exhibit a re-finding pattern, leading to substantial enhancements in the search result quality and relevance.

(2) The performance of CoPS is competitive with, and in some instances even exceeds, traditional fine-tuned personalized search models. This substantial narrowing of the disparity between zero-shot baseline and fine-tuned baseline is attributable to the robust user modeling capabilities of LLMs. Moreover, CoPS adeptly get over the challenges posed by acquiring high-quality supervised data in search scenarios, thereby addressing a critical bottleneck inherent in neural personalized search methods.

### 5.2 Effect of Cognitive Memory Mechanism

In this section, we conduct a comprehensive investigation into the impact of cognitive memory mechanism by using ablation studies and assessing how varying history lengths affect its performance.

*5.2.1 Ablation Studies.* To evaluate the contribution of different memory units on ranking performance and query latency, we conducted ablation studies by systematically removing each memory unit from the model. The results are summarized in Table 2. Our findings reveal that omitting any memory unit results in decreased performance. Notably, the removal of long-term explicit memory exhibits the most substantial negative impact, suggesting that the user's long-term interests plays a crucial role in personalization. In addition, we observed that when the sensory memory is removed, there is a notable increase in query latency and a decline in results. This finding highlights the importance of sensory memory in both the efficacy and efficiency of the model, indicating that the utilization of re-finding behavior is a simple yet effective approach for personalization.

*5.2.2 Effect of Different History Lengths.* To examine the influence of varying user history lengths on the model's performance, we retained different proportions of the user's most recent interactions at intervals in increments of 10% for experiments. As illustrated in

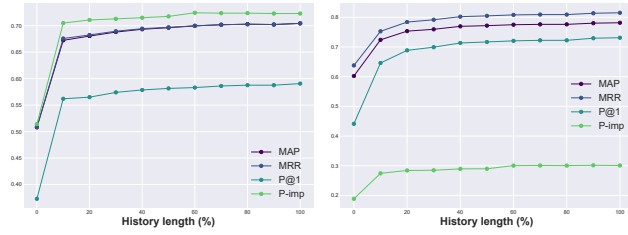

| (a) AOL dataset | (b) Commercial dataset |

**Figure 3: Performance of different history lengths.**

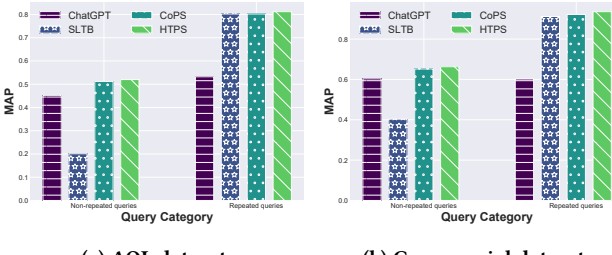

| (a) AOL dataset | (b) Commercial dataset |

**Figure 4: Performance on different query sets.**

Figure 3, our results indicate that a longer user history enhances the model's performance. Notably, the most recent user behavior stands out as the most influential factor, suggesting its importance in personalization. As the temporal gap between interactions widens, the influence of historical behaviors on current search result personalization lessens.

### 5.3 Exploration of the Role of LLM

In this section, we conduct a comprehensive analysis to explore the essential functions of LLMs for personalized search. Specifically, we investigate three core functions of LLMs: query re-writing, user profile retrieval, and user modeling. The goal is to gain a deeper understanding of their importance and influence on the personalized search framework. To this end, we introduce three variants for comparison: (a) a setup where all LLM functions are entirely omitted (-Remove); (b) an approach where the LLM functions are replaced with a random sample strategy (-Random); and (c) a version where ChatGPT is replaced by the less potent language model, Vicuna-7B [33] (-Vicuna).

The results, as presented in Table 3, demonstrate that LLMs play a pivotal role in each step of the personalized search pipeline, with particular emphasis on user profile retrieval and user modeling. When we switch out the LLMs for the less advanced Vicuna-7B model, there is a noticeable drop in the performance metrics. These findings underscore the complexity of user modeling in personalized search tasks and highlight ChatGPT as a potent solution to address this challenging task.

### 5.4 Performance on Different Query Sets

We partition the test queries into two categories: repeated and non-repeated queries. Repeated queries benefit from readily available user click data on the same queries in the past, facilitating the inference of user behaviors. However, non-repeated queries lack such direct references, leading to a dearth of information for

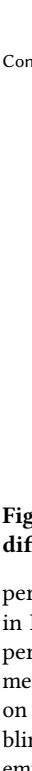
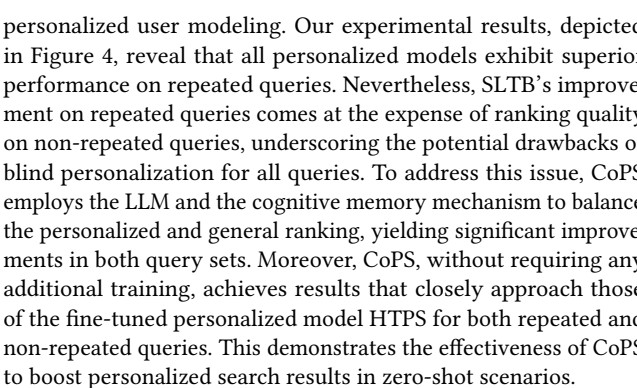

**Figure 5: Analysis of fine-tuning and inference efficiency on different models.**

personalized user modeling. Our experimental results, depicted in Figure 4, reveal that all personalized models exhibit superior performance on repeated queries. Nevertheless, SLTB's improvement on repeated queries comes at the expense of ranking quality on non-repeated queries, underscoring the potential drawbacks of blind personalization for all queries. To address this issue, CoPS employs the LLM and the cognitive memory mechanism to balance the personalized and general ranking, yielding significant improvements in both query sets. Moreover, CoPS, without requiring any additional training, achieves results that closely approach those of the fine-tuned personalized model HTPS for both repeated and non-repeated queries. This demonstrates the effectiveness of CoPS to boost personalized search results in zero-shot scenarios.

## 5.5 Analysis of Efficiency

In personalized search, the user modeling step often consumes a significant amount of time, making efficiency a crucial consideration. Specifically, we break down efficiency into two aspects: fine-tuning efficiency and inference efficiency. The former refers to the number of hours spent on training the model for downstream tasks, while the latter denotes the query latency during actual use. To showcase the efficiency of CoPS more comprehensively, we test the inference efficiency of invoking the API and locally deployment (6B LLM).

As depicted in Figure 5, the time required for training and inference in fine-tuned models exhibits a direct relationship. Among them, the SLTB model stands out for its high efficiency, although it showcases the poorest performance. The HRNN and RPMN models, leveraging RNN structures, manage to enhance the model's complexity and achieve better results simultaneously. By replacing the RNN structure with a transformer structure, the HTPS model not only attains improved performance but also elevates efficiency. Concerning zero-shot models, our proposed CoPS, without utilizing any fine-tuning data, achieves inference efficiency and ranking performance comparable to that of fine-tuned models. Moreover, local deployment of the model can also mitigate the impact on inference speed caused by network latency.

## 5.6 Case Study

To provide a more intuitive demonstration of the workings of our proposed CoPS mechanism, we conduct a case study to observe

**Table 4: The case study of CoPS on the query "*Maybelline new yorky*". The same color indicates that CoPS is helpful for correctly matching the ground-truth document.**

| Query | Maybelline new yorky |
|---|---|
| Sensory Response | No re-finding data found |
| Query Re-writing | *Maybelline New York make up* |
| User Profile Retrieval | ***Explicit Memory Retrieval***
*-Shoes: sandals, designer shoes*
*-Cosmetics Products: MAC, Loreal Paris Hair*
*-Salon Services: Killeen, Texas, hair styling*
***Implicit Memory Retrieval***
*-Gender: Female*
*-Age: teens to middle-aged*
*-Social Image: Beauty Enthusiast, Fashion* |
| User Modeling | Fashion trends featuring Maybelline New York cosmetics and make up products |
| Ground-truth document | *Make up products, Make up tips, and fashion trends maybelline new york* |

the role of the LLM throughout the pipeline, which includes query re-writing, user profile retrieval, and user modeling. We have color-coded the content generated by the LLM that aids in personalized ranking for clearer distinction.

As shown in Table 4, the input query was "Maybelline new yorky", which contains a spelling error. CoPS initially channels it through the sensory memory for re-finding identification. When the sensory response indicates "No re-finding data found", the query is directed to the query re-writing module. Here, the LLM corrects and expands the initial query and subsequently retrieves a related user profile from the long-term memory. Finally, using this combined information, the LLM performs user modeling and deduces the user's personalized search intent as "Fashion trends featuring Maybelline New York cosmetics and makeup products". This refined query aligns much more closely with the ground-truth document than the original one, ensuring that the document is ranked higher in personalized search results.

## 6 CONCLUSION

In conclusion, we have presented CoPS which combines LLMs and a cognitive memory mechanism to enhance user modeling and improve personalized search results. To tackle the pervasive challenges of data sparsity and lengthy, noisy user interactions, our design adopts external memory components, drawing inspiration from human brain's memory system. CoPS integrates three essential components: the sensory memory, working memory, and long-term memory. These memory units allow the model to efficiently store and retrieve user interactions, enabling quick responses to re-finding behaviors and constructing a comprehensive user profile for effective query intent modeling. The integration of LLMs into personalized search opens up new avenues for future research. A direction worthy of research is the privacy protection and security issues in the fusion of personal data and LLMs, promoting the birth of a reliable personal intelligent information assistant.

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
