# OpenReview forum: "Cognitive Personalized Search Integrating Large Language Models with an Efficient Memory Mechanism"
_ACM.org/TheWebConf/2024/Conference — TheWebConf24 Oral_

### Official Review · Reviewer_BUEJ · 2023-11-07

**Novelty:** 5
**Technical Quality:** 4

**Review:**

The paper proposed an LLM based method to summarize, retrieve and use personal information for personalized search. The three stage memory system makes sense to me, and the experiment result seems to be good.

One part which I don’t quite agree with the authors is “analysis of efficiency”. In this section, the authors seem to claim the proposed method is efficient. But if I refer back to Section 3.2, it seems like, for each search query, the proposed method needs to ask an LLM a few reasoning questions. LLM inference is usually way heavier than that of retrieval / re-ranking. So to me, the proposed method costs a lot of resources, and may not be applicable to real use cases (the latency of doing multiple LLM inferences could just be too long for the search use case).

**Questions:**

Any latency / resource usage analysis on the proposed method?

**Ethics Review Description:**

It is not clear if the paper has special treatments to sensitive personal topics like health, etc.

**Ethics Review Flag:**

Yes

**Reviewer Confidence:**

4: The reviewer is certain that the evaluation is correct and very familiar with the relevant literature

**Scope:**

4: The work is relevant to the Web and to the track, and is of broad interest to the community

---

### Official Review · Reviewer_aYVe · 2023-11-22

**Novelty:** 5
**Technical Quality:** 4

**Review:**

I find the argument for why the new model is needed very vague: "existing personalisation models encounter significant challenges stemming from data sparsities and the complexities associated with user history." What these challenges are are only alluded to in Section 1, and without any references nor empirical justification (see lines 77-92).

So what we are left with is the neurologically inspired framework for personalisation (which uses LLMs), which seems to work. But the motivations and positioning for why the framework is better than others for this task is entirely lacking. Indeed, for the "external" baselines in Table 1, all we are told is that is usually better or comparable to.

The ablation study in Table 2 is useful, but it only reduced effectiveness as far as 0.62 on the first dataset, which is still higher than most of the personalises search baselines.

Overall, while I like the architecture, I think the authors should have placed more emphasis on positioning this work within the literature much better, to identify the key insights that the model brings, rather than just its neurologically inspired.

Section 3.2.1:
- what does the square brackets mean in the prompt mean, eg [Query]?

Section 3.3.1:
- The paper appears to confuse the notions of Ranker and Reranker. You have BM25 which is conventionally a Ranker/Retriever. But Vector-based rankers (BERT) and LLM-based ranker can only be a REranker, as they take candidate documents. How are these candidates identified?

Section 2.1:
 - line 180: what is the sequential data leveraged by Ge et al.? The sentence is vague

Section 4.2:
 - what is "IF-IDF" weighting - typo?

**Questions:**

- What is your candidate ranker?
- How did you verify the performances of the external baselines? Did any of those report performances on the AOL dataset?

**Ethics Review Description:**

I'm not clear on the ethics of using the AOL dataset, which has been withdrawn.

**Ethics Review Flag:**

Yes

**Reviewer Confidence:**

3: The reviewer is confident but not certain that the evaluation is correct

**Scope:**

4: The work is relevant to the Web and to the track, and is of broad interest to the community

---

### Official Review · Reviewer_KAmT · 2023-11-23

**Novelty:** 5
**Technical Quality:** 6

**Review:**

The paper introduces CoPS, a Cognitive system for Personalized Search. CoPS is proposed as an alternative to typical personalized search and user modeling systems that rely on large amounts of personal user data. CoPS is inspired by human memory, and is composed of a sensory memory (i.e. a sort of cache), a working memory dealing with in-session information, a long-term memory storing user profiles and a re-ranker model that produces the final list of results. Leveraging LLMs, CoPS only relies on high-level text-based user profiles and information that are derived from fine-grained user interactions. This approach allows to obtain best-in-class performance in low-data scenarios. The system is completely unsupervised.

Pros:
- The paper is well written and easy to read.
-The contribution is original and the experimental setup is sound.
-The ablation studies investigate the importance of the subcomponents of the system.

Cons:
-The main limitation of this paper is that it heavily relies on a large-scale LLM (ChatGPT) for almost all the steps of the personalized search process. At scale, this would imply prohibitive costs. Replacing ChatGPT with a smaller model Vicuna seems to degrade performance (Tab. 3). How would you run such a system at scale?

Other Comments:
- The authors mention that one of the advantages of such a system would be an enhanced user privacy, through a local deployment of LLMs on edge. This argument should be better elaborated: which components would be running on device and which parts on the server side to preserve user privacy?

**Questions:**

See Cons and Other Comments

**Reviewer Confidence:**

4: The reviewer is certain that the evaluation is correct and very familiar with the relevant literature

**Scope:**

4: The work is relevant to the Web and to the track, and is of broad interest to the community

---

### Official Review · Reviewer_j9VL · 2023-11-29

**Novelty:** 6
**Technical Quality:** 7

**Review:**

Personalized search methods based on deep learning rely on a large amount of training data, making them susceptible to the challenge of data sparsity. This paper uses LLM and Cognitive memory mechanisms  to enhance user modeling and user search experience. CoPS effectively processes new queries using three methods: identifying rediscovery behavior, constructing user profiles containing relevant historical information, and ranking documents based on personalized query intentions.
The organizational structure of this article is reasonable, the expression is clear, and the experimental design is complete.
This article explores the role of LLM in personalized retrieval.

**Questions:**

The author needs to further explain the specific performance of the model in querying data sparsity.

**Reviewer Confidence:**

4: The reviewer is certain that the evaluation is correct and very familiar with the relevant literature

**Scope:**

4: The work is relevant to the Web and to the track, and is of broad interest to the community

---

### Official Review · Reviewer_ymJf · 2023-12-01

**Novelty:** 5
**Technical Quality:** 4

**Review:**

This paper focuses on how to conduct cognitive personalized search by integrating LLMs with an efficient memory mechanism. The topic is interesting, and the proposed model is easy to follow. Various experiments are conducted to show the effectiveness of the designed model.

Some Strengths:
1. The author's motivation is clearly introduced, addressing an important issue in the field.
2. The workflow of the work is detailed introduced, and the methods section of the paper is easy to understand.
3. The author adopted various baseline models for comparison and conducted a series of experiments to verify the effectiveness of the proposed method.

However, there are also some weaknesses:
1. The proposed model leverages the capabilities of LLMs and considers a lot of user information that is usually hard to obtain in common scenarios. Its effectiveness is slightly weaker than the personalized search method HTPS. What improvements do the authors think could be made in the future to enhance the practicality of the method?
2. The paper does not clearly explain the evaluation setup, particularly how many document candidates are involved in the ranking for each query, which is a very important parameter setting. Additionally, the number of users that the model can evaluate is relatively small.
3. Is the proposed strategy potentially strongly tied to ChatGPT? For instance, would it be unable to support smaller models? The author also does not provide the average input token length.

**Questions:**

1. Compared to ChatGPT, what kind of data inclusion would yield the maximum improvement? Currently, Table 2's ablation study only shows that all are helpful.
2. Please respond to some of the issues mentioned in the above weaknesses section.

**Ethics Review Description:**

No ethical issues

**Reviewer Confidence:**

3: The reviewer is confident but not certain that the evaluation is correct

**Scope:**

4: The work is relevant to the Web and to the track, and is of broad interest to the community

---

### Decision · Program_Chairs · 2024-01-22

**Decision:**

Accept (Oral)

**Comment:**

This paper presents the use of cognitive memory mechanisms and LLMs for personalized search.

 The reviewers appreciated the research, but also flagged some concerns about the motivation (e.g., why is this neurologically inspired approach needed?), the results compared with HTPS, the incomplete description of the evaluation setup, and the practicalities (concerns about scalability of using LLMs for this setup). The rebuttals from authors on these issues and others were largely satisfactory and could/should be integrated into a revised version of the paper.

 I lean toward acceptance of this submission, especially given the addressability of the reviewer feedback, the consistently positive scores across the 5 reviewers (especially high in novelty), and a presence of a champion (reviewer j9VL) who gives near-max scores.

 In making edits, the authors should clarify in the paper that although the performance is lower than HTPS, that there are advantages in using their solution vs. HTPS (e.g., no fine-tuning / zero-shot) and clarify why that is important given that it's a one-time task (vs. inference time, which is incurred every query and is higher for CoPS than HTPS per Figure 1).